# Translational Research and Therapies for Neuroprotection and Regeneration of the Optic Nerve and Retina: A Narrative Review

**DOI:** 10.3390/ijms251910485

**Published:** 2024-09-29

**Authors:** Toshiyuki Oshitari

**Affiliations:** 1Department of Ophthalmology and Visual Science, Chiba University Graduate School of Medicine, Inohana 1-8-1, Chuo-ku, Chiba 260-8670, Japan; tarii@aol.com; Tel.: +81-43-226-2124; Fax: +81-43-224-4162; 2Department of Ophthalmology, International University of Health and Welfare School of Medicine, 4-3 Kozunomori, Narita 286-8686, Japan

**Keywords:** neuroprotection, regeneration, retinal diseases, optic nerve, glaucoma, retinitis pigmentosa, diabetic retinopathy, age-related macular degeneration, gene-agnostic therapy, neuroprotectants

## Abstract

Most retinal and optic nerve diseases pose significant threats to vision, primarily due to irreversible retinal neuronal cell death, a permanent change, which is a critical factor in their pathogenesis. Conditions such as glaucoma, retinitis pigmentosa, diabetic retinopathy, and age-related macular degeneration are the top four leading causes of blindness among the elderly in Japan. While standard treatments—including reduction in intraocular pressure, anti-vascular endothelial growth factor therapies, and retinal photocoagulation—can partially delay disease progression, their therapeutic effects remain limited. To address these shortcomings, a range of neuroprotective and regenerative therapies, aimed at preventing retinal neuronal cell loss, have been extensively studied and increasingly integrated into clinical practice over the last two decades. Several of these neuroprotective therapies have achieved on-label usage worldwide. This narrative review introduces several neuroprotective and regenerative therapies for retinal and optic nerve diseases that have been successfully translated into clinical practice, providing foundational knowledge and success stories that serve as valuable references for researchers in the field.

## 1. Introduction

A recent nationwide survey of visual impairment in individuals aged ≥18 years in Japan indicates that glaucoma is the most prevalent causative disease, followed by retinitis pigmentosa, diabetic retinopathy (DR), and age-related macular degeneration (AMD) [1]. The top four sight-threatening conditions are retinal and optic nerve diseases, with a total of 12,026 individuals newly defined as visually impaired individuals because of these diseases between 1 April 2019 and 21 March 2020 [1]. One of the primary causes of visual impairment and blindness associated with these retinal optic nerve diseases is neuronal cell death and neurodegeneration, which are integral to their pathogenesis. Neuronal cell death and neurodegeneration are irreversible changes characterized by the loss of neuronal cell bodies, as well as their axons and dendrites. Consequently, once neuronal cell loss and degeneration occur, visual function cannot be recovered through standard therapies for retinal and optic nerve diseases [2,3,4]. Another significant factor contributing to visual impairment and blindness in these conditions is the limited efficacy of standard therapies, which include eye drops for intraocular pressure reduction, glaucoma surgeries, anti-vascular endothelial growth factor (VEGF) therapies, retinal photocoagulation, and vitrectomy. In the case of retinitis pigmentosa, currently, the only available treatment option is voretigene neparvovec (Luxturna) [5].

To address the limitations of standard therapies, extensive research into neuroprotective and regenerative therapies aimed at preventing the progression of retinal and optic nerve diseases have been performed over the past two decades. Some of these therapeutic strategies have successfully translated into the clinical trials and practice [2,6]. This narrative review introduces several neuroprotective and regenerative therapies aimed at preventing the progression of retinal and optic nerve diseases, including glaucoma, retinitis pigmentosa, DR, and AMD. Given the rapid advancements in medical science in this area, it is essential to continuously update knowledge based on the latest scientific literature. This narrative review aims to enhance understanding of neuroprotective and regenerative strategies for retinal and optic nerve diseases.

## 2. Glaucoma

Glaucoma is a progressive optic neuropathy characterized by the irreversible degeneration of retinal ganglion cells (RGCs) and their axons. By 2040, the predicted prevalence of glaucoma is expected to exceed 112 million individuals worldwide [7]. The main cause of glaucomatous optic neuropathy is increased intraocular pressure (IOP), and the only established treatment option for delaying its progression is IOP reduction [7]. However, the therapeutic effects of IOP reduction are evidently limited in preventing glaucoma-related visual impairment and blindness, particularly in Japan [1]. One potential reason for the limitations of standard treatments for glaucoma in Japan is that 92% of those with primary open-angle glaucoma (POAG) have normal tension glaucoma; thus, the therapeutic effect of IOP reduction is relatively limited compared to that seen in high tension glaucoma [8,9]. This indicates that IOP-independent pathways may play a role in the pathogenesis of normal tension glaucoma. Additionally, RGC death and axonal degeneration are related to the pathogenesis of glaucomatous optic neuropathy. As RGC loss is irreversible, there is an urgent need to establish neuroprotective and regenerative therapies to prevent RGC degeneration in patients with glaucoma.

### 2.1. Molecular Pathogenesis in Glaucoma

Although the precise mechanisms underlying RGC loss in glaucoma remains unclear, RGC apoptosis is partially implicated in the visual decline in patients with glaucoma [10,11,12]. Research on the apoptotic cell death mechanisms in the human retina affected by glaucoma is limited, primarily because glaucoma is a chronic condition that complicates the identification of key apoptotic factors. However, Tezel et al. successfully demonstrated that the stress-activated protein kinase, c-Jun N-terminal kinase (JNK), is expressed in RGCs of individuals with glaucoma [13]. Activated phosphorylated JNK serves to activate c-Jun, which then mediates pro-apoptotic gene expression; the upregulation of the JNK/c-Jun pathway has been correlated with RGC death in both animal models and human cases of glaucoma [13,14,15]. Although JNK is a target for neuroprotective interventions [16], c-Jun can be activated in chronic diseases, such as glaucoma, even in the absence of JNK2 and JNK3 [17]. Thus, the chronic effect of JNK inhibitors warrant caution when used in chronic conditions, including glaucoma [17]. Currently, no JNK inhibitors have been tested in clinical practice for patients with glaucoma.

Tumor necrosis factor-alpha (TNF-α) is a pro-inflammatory cytokine that binds to TNF receptor 1 (TNFR1) and 2 (TNFR2), resulting in RGC death after optic nerve crush [18] and ocular hypertension [19]. Previous studies indicate that TNF-α is upregulated in the retinas of individuals with glaucoma [20,21]. Yuan et al. indicate that in eyes with severe glaucomatous optic neuropathy, TNF-α and TNFR1 are expressed in microglia, while TNFR1 is also expressed in RGC axons. Thus, damaged RGCs may serve as direct targets of TNF-α in severe glaucoma [21]. A recent study indicates that TNF-α and transforming growth factor-β2 (TGF-β2) in the aqueous humor are higher in glaucoma-affected eyes compered to control cataract-affected eyes [22]. Another recent study suggests that the level of TNF-α in the aqueous humor of eyes with normal tension glaucoma is positively associated with the presence of central scotomas [23]. TNFR1 recruits TNF receptor 1-associated death domain protein (TRADD), subsequently leading to the activations of caspases-8 and -3 [3]. In fact, in human eyes with glaucoma, TNF-α, TNFR1, and TRADD are upregulated from protein samples, and active forms of caspases-3, -8, -9, and -12 are detected in RGCs [24]. Notably, TNFR2 is not detected in human donor samples from glaucomatous eyes [24]. The proteomic analysis of glaucomatous human retinas indicates that TNF-α/TNFR1 signaling plays a significant role in the pathogenesis of glaucomatous optic neuropathy [24]. Although several studies have demonstrated the therapeutic effect of TNF inhibition for optic nerve injury and glaucoma [25,26,27], clinical studies evaluating the efficacy of TNF inhibitors for primary or secondary glaucoma remain limited [28,29]. Although not a direct TNF inhibitor, bupropion—an antidepressant used to aid smoking cessation—is a norepinephrine–dopamine reuptake inhibitor [30]. Its properties may suppress TNF production [31]. A large cohort study (n = 638,481) found that users of bupropion showed a significant hazard with the development of open-angle glaucoma compared to non-users, after adjusting for potential confounding factors [32]. However, as this was a retrospective study, further prospective randomized control studies are necessary to establish whether bupropion can serve as a novel therapeutic option for open-angle glaucoma. It is important to note that bupropion is not suitable for angle closure glaucoma and is still used off-label in Japan. Puerarin (7,4’-dihtdroxy-8-C-glycosylisoflavone), an isoflavone, is found in the dried root of *Peuraria montana* var. *thromsonii* (Benth.) M.R.Almeida (*syn. Pueraria thromsonii* Benth.) or *Peuraria montana* var. *lobata* (Wild.) Maesen & S.M.Almeida ex Sanjappa & Predeep (syn. *Pueraria. Thromsonii* Benth.) [33]. Puerarin exhibits anti-inflammatory, anti-oxidative, and anti-apoptotic effects by regulating various inflammatory- and apoptosis-related factors, including TNF-α, interleukin-1beta (IL-1β), intercellular adhesion molecule-1 (ICAM-1), nuclear factor kappa-B (NF-κB), mitogen-activated protein kinases (MAPKs), and B-cell lymphoma-2 (Bcl-2) [34]. However, only a limited number of studies in the Chinese-language literature have reported on the effects of puerarin in patients with glaucoma [34], highlighting the need for further clinical investigations to confirm its therapeutic potential in preventing the progression of glaucomatous optic neuropathy.

### 2.2. Memantine Failure

Excitotoxicity is a major stressor that induces neuronal cell death in various neurological conditions, including glaucoma. Glutamate-mediated transmission plays a pivotal role in initiating excitotoxic cell death [35]. Excessive glutamate activates N-methyl-D-aspartate (NMDA) receptors, which result in increasing Ca^2+^ influx and the activation of cell death cascade in neurons [35]. Numerous studies of retinal ischemia and glaucoma models have demonstrated that intravitreal NMDA injections induce retinal neuronal cell death and that NMDA receptors and/or downstream pathways’ inhibition shows neuroprotective effects against excitotoxic injury [36,37,38,39,40,41,42,43]. However, the results of glutamate levels in the vitreous of glaucoma models are conflicting, hindering the development of effective anti-excitotoxic therapies for glaucoma [44,45,46,47]. Memantine, an open-channel-blocking NMDA antagonist, does not affect normal transmission but inhibits the overexpression of NMDA receptors [48,49]. Several studies indicate that memantine ameliorates excitotoxic damage in glaucoma models [37,50,51]. Because memantine has already been approved for the treatment of Alzheimer’s and Parkinson’s diseases, two randomized, double-masked, placebo-controlled, parallel-group, and multicenter studies lasting 48 months were performed to evaluate the neuroprotective effect of oral memantine in open-angle glaucoma [52]. Despite widespread anticipation for positive outcomes, oral memantine did not provide the expected neuroprotective effect for patients with glaucoma [52]. Given the substantial sample size of 2298 participants, most clinicians and researchers in the field of glaucoma were disappointed by these the negative results. Consequently, enthusiasm for the development of neuroprotective therapies for glaucoma appears to have waned for the time being.

### 2.3. Neuroprotectants for Glaucoma

Citicoline, also known as cytidine-5’-diphosphocholine, or CDP-choline, is an intermediate product in the synthesis of phosphatidylcholine, which is a predominant phospholipid in neuronal membranes [53]. Exogenous citicoline is rapidly hydrolyzed into cytidine-5’-monophosphate and phosphocholine by phosphodiesterases in the cell membrane before being absorbed by neuronal cells in the brain and retina [54,55]. Although the precise mechanisms by which citicoline exerts it protective effects remain to be elucidated, exogenous citicoline is thought to integrate into cell membranes, which accelerates the synthesis of membrane phospholipids and stabilize intracellular conditions after damage [54,55]. Because of its favorable safety profile and the high bioavailability, citicoline has long been used clinically as a neuroprotectant for various conditions, including Parkinson’s disease, Alzheimer disease, stroke, and brain injury [54,55]. Oshitari et al. were the first to demonstrate the neuroprotective and regenerative effect of citicoline on damaged retinal neurons in culture [56]. Subsequent studies have demonstrated the neuroprotective effect of citicoline on damaged retinal neurons, including RGCs, both in vitro and in vivo [57,58,59,60,61,62,63]. The neuroprotective effects of citicoline correlate with the suppression of active forms of caspases-9 and -3 [59], Bcl-2 [58], and phosphorylated JNK [62]. Additionally, citicoline may reduce the activity of phospholipase A_2_, which is involved in mitochondrial membrane degradation [64]. Taken together, citicoline may serve as a mitochondrial stabilizer with protective effects.

Several clinical studies indicate that citicoline eye drops or oral administration can slow disease progression or improve retinal function in patients with glaucoma [65,66,67]. Furthermore, a recent three-year randomized placebo-controlled study indicates that citicoline eye drops reduced disease progression in patients with progressive glaucoma whose intraocular pressure (IOP) was 18 mmHg or less [68]. Finally, an international, multicenter, randomized, placebo-controlled, and cross-over study also demonstrated that oral administration of citicoline improved the vision-related quality of life in patients with glaucoma [69]. It is anticipated that oral administration or topical instillation of citicoline will soon be formally approved for use in patients with glaucoma worldwide. In fact, citicoline eye drops have already been registered in many European Union (EU) countries (Italy, France, Spain, etc.) as well as non-EU countries (Iran, Kazakhstan, Uzbekistan, etc.). In contrast to memantine, citicoline appears to be on a promising path toward successful integration as a neuroprotective therapy for glaucomatous optic neuropathy.

Brimonidine is an IOP-lowering agent that is clinically used to treat patients with glaucoma. It is a specific *α*2-adrenoreceptor agonist and has a neuroprotective effect on RGCs independent of its IOP-lowering effect [70]. Although the precise mechanism by which brimonidine exerts its neuroprotective effects is not entirely understood, it can modulate NMDA receptor function and protect RGCs against excitotoxicity [71]. Brimonidine can facilitate inhibitory post synaptic activity of RGCs and reduce excitotoxic damage during the pathological process of glaucoma [72]. A randomized, double-masked, and multicenter clinical trial, known as the Low-Pressure Glaucoma Treatment Study, found that the 0.2%-brimonidine-treated group was less likely to experience visual field progression compared to the 0.5%-timolol-treated group [73]. Because the mean treated IOP was similar for both groups at all time points, the reduced disease progression observed in the 0.2%-brimonidine group highlights its potential neuroprotective effects [73]. However, in the Low-Pressure Glaucoma Treatment Study, the rate of disease progression in the 0.5%-timolol group was higher than that in the previous non-treatment group [74]. In addition, the frequency of ocular allergy in the 0.2% brimonidine group was significantly high, indicating the need for further study to examine the neuroprotective effect of brimonidine in patients with glaucoma [75]. In fact, a lower dose of brimonidine (0.1%) is clinically used in Japan, as ocular allergy is a major reason for discontinuing the drug. It remains uncertain whether long-term use of 0.2% brimonidine is tolerable for patients with chronic glaucoma.

Axonal transport blockade, coupled with neurotrophic factor deprivation, is believed to be one of the major pathological factors for the development and progression of glaucomatous optic neuropathy [76,77,78]. Furthermore, neurotrophic factors in the tears and the serum of patients with glaucoma are lower than those in the controls [79,80]. Thus, neurotrophic supplementation therapies, such as nerve growth factor (NGF), brain-derived neurotrophic factor (BDNF), or ciliary neurotrophic factor (CNTF), have been studied for preventing the progression of glaucoma [81,82,83]. Some neurotrophic factor supplementation therapies have been translated into clinical studies. Recombinant human NGF (rhNGF) eye drops were used in examining the safety, tolerability, and efficacy in patients with glaucoma [84]. This Phase 1b study indicated that topical rhNGF showed no adverse events but also lacked statistically significant short-term neuroenhancement [84]. Further analysis of the efficacy via a neuroprotection trial may be performed in the future. Similarly, Phase 1 NT-501 CNTF implants were tested for safety and efficacy in patients with glaucoma [85]; it was found that this implant was safe and well tolerated in patients with glaucoma and both structural and functional improvement were observed in implanted eyes [85]. Consequently, a randomized Phase II trial is currently underway (NCT02862938).

Oxidative stress contributes to neuronal cell death and is associated with the pathogenesis of various neurodegenerative diseases [86]. Increased production of reactive oxygen species (ROS) and decreased levels of antioxidative components are central contributors to oxidative stress. Conversely, reducing ROS production and/or increasing antioxidants may offer neuroprotective benefits against neurodegenerative diseases. Malondialdehyde (MDA) is considered a pro-oxidant biomarker, and its levels are significantly elevated in the serum and/or aqueous humor of patients with POAG compared to controls [87,88]. Additionally, 8-hydroxy-2’-deoxyguanosine (8-OHdG), a pro-oxidant form, a marker of free-radical-induced oxidative damage, was significantly increased in patients with POAG compared to controls [89,90]. Contrastingly, glutathione peroxidase (GPx), superoxide dismutase (SOD), and catalase (CAT) are anti-oxidant biomarkers, with the total antioxidant capacity (TAC) being a reliable marker of the complete antioxidant status [91]. The TAC is significantly decreased in patients with glaucoma [87,91,92], suggesting that oxidative stress is involved in the pathogenesis of glaucoma and that antioxidant therapies may represent viable treatment options for patients with glaucoma. Coenzyme Q10 (CoQ10) functions as a cofactor in the electron transport chain and contributes to stabilizing mitochondrial membrane potential, which subsequently enhances ATP formation and reduces ROS production. CoQ10 has been shown to ameliorate ischemic and excitotoxic damage in RGCs [93,94]. CoQ10 and vitamin E eye drops (COQUN) improved pattern electroretinograms and visual-evoked potentials in patients with POAG [95]. However, this study does not show a neuroprotective effect of CoQ10, and thus, further clinical studies are needed to evaluate the protective effect of CoQ10 in patients with glaucoma. 

Nicotinamide (NAD) is a precursor of nicotinamide adenine dinucleotide (NAD^+^) and plays a key role in the glycolysis pathway, contributing to ATP generation [96]. NAD^+^ levels decline with aging, leading to mitochondrial dysfunction, reduced ATP production, and increased ROS production [97]. On the contrary, NAD^+^ supplementation can ameliorate stress-induced mitochondrial dysfunction, leading to increased ATP production and reduced ROS production in damaged neurons. In fact, nicotinamide supplementation has been shown to sustain mitochondrial function and protect against glaucomatous RGC damage in aged [98] and inherited glaucoma mouse models [99]. A small cohort study indicates that the plasma levels of NAD may be lower in patients with glaucoma than those in controls [100]. A recent crossover, double-masked, and randomized clinical trial indicates that NAD supplementation improves inner retinal function determined by a photopic-negative response in patients with glaucoma [101]. A recent Phase 2 randomized clinical trial suggests that a combination of NAD and pyruvate significantly improve visual function determined by visual field tests in patients with glaucoma [102]. A recent ongoing randomized, double-blinded, placebo-controlled, parallel-group, and multi-center study is investigating whether nicotinamide riboside, an NAD precursor, is effective in preventing the progression of optic nerve degeneration in patients with glaucoma for 2 years (Chinese Clinical Trial Registry 1900021998) [103]. Taken together, supplementation of NAD and/or its precursors may represent promising neuroprotective therapies for patients with glaucoma, independent of IOP reduction.

## 3. Age-Related Macular Degeneration

### 3.1. Pathogenesis in AMD

AMD is an acquired condition that causes central visual impairment. The pathogenesis of macular degeneration is associated with several environmental, nutritional, aging, and genetic factors [104]. The prevalence of AMD was estimated to be approximately 196 million in 2020 and expected to increase to 288 million worldwide by 2045 [105]. This condition is notably the most prevalent vision-threatening disease in Eastern countries. Current standard therapies for neovascular AMD involve intravitreal injections of anti-VEGF agents. However, therapies for preventing the progression of dry AMD and geographic atrophy (GA) are still under investigation. While the precise mechanisms underlying the development and progression of AMD remain unclear, age and smoking are recognized as significant environmental risk factors [106,107]. Other risk factors are greater body mass index [108], White race [109], and genetic risk factors [110]. Especially, genetic risk factors may provide insights into the pathogenesis of AMD. For example, variants in the complement gene regions, such as the *complement factor H (CFH)* gene [111,112,113], *complement component 2 and factor B (C2/CFB)* gene [114], *complement factor I (CFI)* [112], and *complement component 3 (C3)* gene [112,115], are found to be associated with the pathogenesis of AMD. Gene variants, apart from the complement genes, such as *age-related maculopathy susceptibility 2 and high-temperature requirement A serine peptidase 1 (ARMS2/HTRA1)* genes [113,116,117,118], lipid metabolism genes, and *APOE* genes [119], are also risk factors for AMD.

Photoreceptors and retinal pigment epithelium (RPE) are the primary affected cells in AMD, while RPE is the most susceptible to damage, leading to secondary photoreceptor degeneration. The area of the macula is continuously exposed to oxidative stress because of its high metabolic activity and continuous ROS production [120]. Environmental risk factors, such as excessive light exposure, smoking, poor dietary habits, and systemic risk factors, including obesity, hypertension, and arteriosclerosis, may cause increased oxidative stress followed by inflammation in the macular area, resulting in subsequent RPE and photoreceptor degeneration [110,121]. Overproduction of ROS leads to damage to mitochondrial DNA followed by mitochondrial dysfunction and further degeneration of RPE and retinal neuronal degeneration [122]. Consequently, therapeutic targets for preventing the progression of dry AMD and GA include complement pathways, mitochondrial enhancers, anti-oxidants, and neuroprotective and regenerative therapies, such as autologous-induced pluripotent stem cells (iPSCs), bone-marrow-derived stem cells (BMSCs), and retinal implants.

### 3.2. Complement Pathways in GA

Inhibition of complement pathways is a promising therapeutic avenue for preventing the progression of dry AMD and GA. The complement pathways are activated by the following three factors: antibody–antigen complex in the classical pathway, serum lectin binding to mannose residues on pathogens in the lectin pathway, and direct C3b protein binding to microbes in the alternative pathway [123]. The initial cascades of three pathways culminate with C3 activation. Cleavage of C3 generates C3a and C3b followed by C5 activation. C5 acts as a carrier component of the membrane attack complex (MAC), and after cleavage into C5a and C5b, C5b forms the C5b-9 MAC with C6, C7, C8, and C9 (Figure 1). The MAC formation is related to retinal cell apoptosis in various diseases including AMD [123]. Various complement pathway inhibitors have been investigated in clinical trials and two inhibitors, pegcetacoplan (Syfovre) and avacincaptad pegol (Izervay), were approved by the Food and Drug Administration (FDA) for the treatment of GA secondary to AMD. Pegcetacoplan, a pegylated C3 inhibitor peptide, represents the first FDA-approved treatment for GA [124,125]. Recently, two multicenter, randomized, double-masked, and sham-controlled Phase 3 studies (OAKS and DERBY; NCT03525613 and NCT03525600, respectively) lasting 24 months were completed and published in *The Lancet* [125]. A total of 1258 patients with GA secondary to AMD were enrolled. Patients were randomly assigned to intravitreal pegcetacoplan monthly or every other month, or a sham treatment monthly or every other month. The total areas of GA lesions, as measured by fundus autofluorescence imaging, were evaluated. At 24 months, both the monthly and every other month pegcetacoplan-injection groups showed significantly slower growth in GA lesions compared to the sham groups [125]. The studies conclude that pegcetacoplan slows GA lesion growth in patients with GA secondary to AMD [125]. Another complement pathway inhibitor is avacincaptad pegol (Izervay), an anti-C5 aptamer, has also recently received FDA approval for the treatment of GA associated with AMD. A recent international, prospective, randomized, double-masked, sham-controlled, and pivotal Phases 2/3 clinical trial (GATHER1 Study) was completed [126]. In addition, a second confirmatory pivotal, randomized, and double-masked Phase 3 clinical trial was completed, confirming the efficacy and safety of avacincaptad pegol (Izervay) in slowing GA growth (GATHER2 Study; NCT04435366) [127]. Intravitreal administration of avacincaptad pegol (Izervay) at doses of 2 mg and 4 mg showed significant reductions in GA growth in eyes affected by AMD [126]. Monthly administration of 2 mg avacincaptad pegol (Izervay) was well tolerated, and significantly slowed GA growth over 12 months was observed compared to the sham treatment group [127]. These complement pathway inhibitors are expected to be used for the treatment of slowing GA growth globally in the near future. Other promising complement pathway inhibitors, such as an antisense oligonucleotide-targeting human complement factor B gene, IONIS-FB-LRX (NCT03815825), and humanized IgG1 monoclonal antibody against C3, NGM621 (NCT04465955), have been developed, with Phase II studies currently underway. Various complement pathway inhibitors are anticipated to be integrated into clinical practice for the management of GA associated with AMD. Although not all clinical trials reached the primary endpoints [128,129], some complement pathway inhibitors aimed at slowing GA atrophy secondary to AMD are emerging as novel treatment options for preventing neurodegeneration in this condition.

## 4. Retinitis Pigmentosa

Retinitis pigmentosa is a major inherited retinal disease and the second leading cause of blindness in Japan [1]. The prevalence of retinitis pigmentosa is approximately 1/3000 to 1/4000, with an estimated 2.5 million patients worldwide suffering from visual disturbances including night blindness and visual field defects [130]. Various mutations in different types of genes are associated with death of photoreceptors and RPEs in patients with retinitis pigmentosa; however, most patients adhered to the Mendelian laws of inheritance, i.e., autosomal dominant, autosomal recessive, and X linked modes [131]. A recent Japanese large-scale sequencing study identified the following major causative genes: *EYS*, *USH2A*, *RP1L1*, *RHO*, *RP1* and *RPGR* [132]. The most frequent causative genes found to be associated with autosomal dominant, autosomal recessive, and X-linked retinitis pigmentosa are *RHO*, *EYS*, and *RPGR*, respectively [132]. The study concluded that East-Asian-specific variants in these causative genes are a major contributor to retinitis pigmentosa in Japan [132].

Pathogenic genes associated with retinitis pigmentosa are thought to disrupt phototransduction process. Mutant proteins or sustained protein aggregates can trigger an unfolded protein response, leading to metabolic dysfunction, increased oxidative stress, inflammation, activation of cell death pathways, and retinal remodeling [133,134,135,136,137]. Photoreceptor cell death and degeneration followed by retinal remodeling are the final stages of pathogenesis in retinitis pigmentosa. Because photoreceptor cell death is an irreversible change, inhibition of photoreceptor cell death is considered to be a therapeutic strategy. Once photoreceptor cell loss occurs, stem cell transplantation should be considered. Furthermore, optogenetic approaches have been recently developed as gene-agnostic therapies for inherited retinal diseases including retinitis pigmentosa [138]. This section introduces therapeutic approaches for retinitis pigmentosa.

### 4.1. Gene Therapies

Gene therapies are a radical treatment option for inherited retinal diseases such as retinitis pigmentosa. However, these gene therapies should be performed before the occurrence of photoreceptor and RPE cell loss. One gene therapy that has been successfully translated into clinical practice is gene augmentation therapy using a recombinant adeno-associated virus (AAV) for patients with Leber congenital amaurosis (LCA). Gene augmentation therapies are particularly well-suited for autosomal recessive diseases, such as LCA, because the loss of function caused by a mutant allele can be functionally ameliorated by introducing a healthy allele. However, such gene augmentation therapies must be applied before photoreceptor loss. LCA, a subtype of retinitis pigmentosa, is characterized by earlier onset and more rapid progression than typical retinitis pigmentosa [139,140]. The *RPE65* gene, which encodes all-trans retinal ester isomerase, is identified as the most frequently mutated gene in LCA [139,140]. Several basic studies indicate that the recombinant AAV-mediated *RPE65* gene transfer to the retina successfully restores visual function in animal models of LCA [141,142,143,144]. Following a series of Phase 1 clinical trials [145,146,147,148], the first randomized, controlled Phase 3 clinical trial of gene augmentation therapy for *RPE65*-mediated inherited retinal dystrophy using AAV2-hRPE65v2 (voretigene neparvovec; LUXTURNA) was conducted [149]. The study indicates that voretigene neparvovec (LUXTURNA)-mediated *RPE65* gene transfer improves functional vision in *RPE65*-mediated inherited retinal dystrophy [149]. Furthermore, a long-term study indicates that improvements in visual function in patients with biallelic *RPE65*-mediated inherited retinal diseases after subretinal administration of voretigene neparvovec (LUXTURNA) were maintained for up to 3 to 4 years, with no product-related serious adverse events [150]. Voretigene neparvovec (LUXTURNA) received FDA approval as the first gene therapy for a genetic disease in the USA in December 2017, followed by approval in the EU in November 2018 and in Japan in August 2023. However, a recent multicenter retrospective study indicates that 8 of 10 patients (80%) who underwent subretinal injection of voretigene neparvovec (LUXTURNA) developed progressive perifoveal chorioretinal atrophy [151]. Given the high incidence of these adverse events and the progressive nature of atrophy, further research is needed to determine whether this complication is associated with vector-related factors or surgical techniques [151].

### 4.2. CRISPER-Cas9

Recombinant AAV-mediated gene argumentation therapies have become standard therapies; however, AAV vectors are limited to delivering genes of up to 4.7-kilobases. In addition, gene augmentation therapies are suitable for only a small proportion of recessive diseases, necessitating the development of alternative gene therapeutic strategies. In response to this need, a novel genome editing method has been developed, which is called clustered-regularly interspaced short palindromic repeats (CRISPR) CRISPER-associated (Cas)-based genome-editing systems [152,153]. For gene editing, a single guide RNA and the Class 2 type II Cas enzyme, Cas9, are required. Cas9 is an endonuclease that recognizes the appropriate protospacer adjacent motif (PAM) sequence to bind target DNA. Once the PAM sequence is identified, the double-stranded DNA unwinds, allowing the Cas9-associated single guide RNA hybridization with the target matched sequence of DNA. After Cas9 binds with the target DNA guided by the single-stranded RNA, Cas9 cleaves both strands of the target DNA, generating double-strand break and inactivating the target gene via the non-homologous end-joining pathway [152,153]. If a template gene is available, a specific DNA template is inserted in the target region through the homology-directed repair pathway [152,153]. If the CRISPER-Cas9 package is delivered in the retinal cells by using AAV vectors, therapeutic CRISPER-Cas9 genome-editing strategies can be established theoretically. In addition, multiple editing events, such as the excision of mutation sites from the genome while maintaining the open reading frame, can be achieved by delivering multiple guide RNAs at separate sites. Patients with LCA type 10 have the common *IVS26* mutation—an adenine-to-guanine point mutation in intron 26 (c.2991 + 1655A > G) that results in complete inactivation of the *CEP290* gene [154]. A pair of *CEP290*-specific guide RNAs coupled with *Staphylococcus aureus* Cas9 delivered by AAV5 (EDIT-101) can remove the aberrant splice donor and restore normal CEP expression [155]. The functional rescue of 10% of foveal cones is hypothesized to be necessary for achieving observable clinical benefits [156]. The editing efficacy of EDIT-101 has been reported to exceed 10% productive edits [155], and thus, EDIT-101 has been approved by FDA for a Phase 1–2 clinical trial of a gene editing therapy aimed at patients with *CEP290*-associated retinal degeneration [157]. In the study, subretinal injection of EDIT-101 was performed in 12 adults and 2 children with *CEP290*-associated retinal degeneration [157]. No serious adverse events were observed, and meaningful improvements were observed in their photoreceptor function [157]. Although the study supports further investigation into CRISPER-Cas9-meidated therapies for other inherited retinal degeneration [157], potential off-target effects and immune and inflammatory responses against Cas9 cannot be ruled out completely. Further long-term studies are needed to evaluate the safety of CRISPER-Cas9-mediated therapies for inherited retinal diseases.

### 4.3. Neurotrophic Factors for Retinitis Pigmentosa

In the late stage of retinitis pigmentosa, significant loss of rod photoreceptors is observed, as most patients harbor rod-specific gene mutations. Simultaneously, most cones without cone-specific mutations also undergo degeneration owing to the lack of support from rods, resulting in the starvation of cones [158]. Neuroprotective therapies target rescue of secondary degeneration of cone photoreceptors. Among the traditional neuroprotective therapies for retinitis pigmentosa are those employing CNTF-supplemented approaches. Repeated injection of CNTF [159], encapsulated cell-based delivery of CNTF [160], and AAV-mediated CNTF delivery [161] have demonstrated significant protection against photoreceptor cell death in several animal models of retinitis pigmentosa. However, several studies suggest that visual function cannot be improved by CNTF [161,162,163], as it can suppress cone opsin expression, resulting in decreased light sensitivity [163]. In a long-term follow-up study of a multicenter, sham-controlled trial, eyes of patients with retinitis pigmentosa treated with CNTF (delivered via an intraocular encapsulated cell implant) exhibited greater loss of total visual field sensitivity compared to sham-treated eyes. Over 60 to 96 months, the implanted group showed no significant efficacy in visual function and retinal structure [164]. Thus, while CNTF may provide protection for retinal ganglion cells in glaucoma, it may not be suitable for photoreceptor protection in retinitis pigmentosa [85].

Rod-derived cone viability factor (RdCVF) is identified as a trophic factor secreted from rod photoreceptors that supports the survival of cone photoreceptors [165]. RdCVF is encoded by the *NXNL1* gene, which is specifically expressed by photoreceptors and interacts with basigin-1 (BSG1) [165]. BSG1 binds to the glucose transporter 1 (GLUT1), facilitating increased glucose uptake in cone photoreceptors and promoting their survival by stimulating aerobic glycolysis [165]. In addition, the *NXNL1* gene encodes a long chain isoform via alternative splicing, RdCVFL. RdCVFL inhibits the microtube protein *τ* phosphorylation and reduces methionine sulfoxide, which results in performing an anti-oxidant defense. RdCVF protein injections have been demonstrated to increase the number of cone cells and improve electroretinogram (ERG) function [166]. Furthermore, AAV-mediated expression of RdCVF and RdCVFL promotes the survival of cone and rod photoreceptors and enhances retinal function in animal models of retinitis pigmentosa [167]. Recently, a first-in human Phase1/2 clinical trial evaluating a gene-independent, cone-preserving therapy for retinitis pigmentosa using SPVN06, i.e., a human-engineered AAV that delivers RdCVF gene in retinal cells, has commenced at a low dose and is ongoing at a medium dose (NCT05748873). The study will be completed by March 2029 (*Invset Ophthalmol Vis Sci*. 2024, 65, 3091; ARVO e-abstract). RdCVF-mediated cone protection represents an important and promising gene-agonistic therapeutic strategy for most patients with retinitis pigmentosa, and further results and studies are eagerly anticipated.

### 4.4. Cell Replacement Therapies

Retinal cell replacement therapies are particularly promising for patients experiencing significant loss of rod and cone cells, as these therapies often rely on the presence of some residual photoreceptors to deliver or rescue vision. Two primary types of stem cells used for cell transplantation are embryonic stem cells and induced pluripotent stem cells. Both stem cells can differentiate into various retinal cells, allowing for transplantation into the host retinal tissues using techniques such as cell suspension, sheet-like tissues, or organoids. Recently, many cell replacement therapies have been translated into clinical trials. Human embryonic- and iPSC-derived photoreceptor precursors have been transplanted into the animal models of LCA and were found to restore retinal function [168,169]. Human pluripotent stem cell-derived cone transplantation into *rd1* mouse retina showed synapse formation between donor photoreceptors and host bipolar cells, resulting in the restoration of retinal function [170]. Additionally, the first study on transplantation of stem cell-derived organoids—produced by a self-organizing process in human retinas in patients with retinitis pigmentosa—has been published [171]; allogeneic iPSC-derived retinal organoid sheets were transplanted into two patients with retinitis pigmentosa and their safety and the efficacy in preserving visual function were evaluated for two years [171]. No major safety concerns were noted at two years, however, there was no improvement in visual function [171], necessitating further large-scale studies and more detailed assessments of visual function.

### 4.5. Optogenetics

Optogenetic approaches are another avenue of exploration, involving the introduction of light-sensitive molecules such as channelrhodopsin [172] or halorhodopsin [173] into surviving retinal cell types, including RGCs and bipolar cells. This technique allows the perception of light stimuli even after the complete loss of rod and cone cells in advanced stages of retinitis pigmentosa. Given that the size of opsin genes is less than 2 kb, AAV-mediated gene transfer has been effectively employed to deliver these gene into remaining retinal cells. Therefore, optogenetic approaches use AAV2-mediated optogene transfer methods via intravitreal injection. Although several clinical trials of optogenetic approaches are ongoing, only the PIONEER study has been completed and published [174]. In the open-label Phase 1/2a study (NCT03326336), AAV vector encoding channelrhodopsin, ChrimsonR, fused to the red fluorescent protein tdTomato [175], was intravitreally administrated into the worse-seeing eye to target foveal RGCs in patients with retinitis pigmentosa who had a visual acuity limited to light perception [174]. The fusion protein, ChrimsonR-tdTomato, exhibited greater efficacy of ChrimsonR expression compared to ChrimsonR alone in the cell membrane [176]. The patients wore light-stimulating goggles to capture images from the real-world visual events, transforming these into monochromatic projections of local 595-nm light pulses on to the retina [174]. Patients were able to perceive, locate, count, and touch different objects only when using the goggles. The study concluded that the optogenetic stimulation of RGCs by a light-projection system linked to a camera is an option for partially recovering visual function in blind patients with advanced retinitis pigmentosa [174]. Additional ongoing clinical trials related to optogenetic approaches are the RESTORE trial, investigating the safety and efficacy of a virally-carried multicharacteristic opsin in retinitis pigmentosa (NCT04945772); and in Stargardt disease (STARLIGHT trial; NCT05417126); a Phase 1/2 trial to evaluate the safety and efficacy of a recombinant AAV-mediated ChronosFP (BS01) delivery into RGCs in retinitis pigmentosa (NCT04278131); and a study evaluating the safety and tolerability of intravitreal administration of RST-001 in patients with retinitis pigmentosa (NCT02556736). Collectively, these optogenetic approaches represent a promising therapeutic option for patients in the later stages of retinitis pigmentosa.

## 5. Diabetic Retinopathy

DR is a major complication of diabetes and a leading cause of blindness worldwide. A recent report from the International Diabetes Federation Diabetes Atlas indicates that the global prevalence of diabetes in individuals aged 20–79 years is approximately 540 million in 2021 and is projected to rise to 780 million by 2045 [177]. Similarly, the global prevalence of DR is estimated to be approximately 103 million in 2020 and is expected to rise to 160 million by 2045 [178]. Furthermore, approximate 10% of patients with diabetes are believed to have vision-threatening stages of DR, i.e., clinically significant diabetic macular edema and proliferative DR [179]. DR is characterized by tissue-specific neurovascular impairment affecting the intricate interdependence among neurons, glial cells, and vascular cells, together forming the neurovascular unit [180]. Thus, ideal therapeutic approaches for preventing the development and progression of DR should focus on protecting all cell types within the neurovascular unit. While various clinical trials investigating treatments for diabetic macular edema have been conducted, this section only focuses on the therapeutic approaches aimed at preventing the onset and the progression of early DR, excluding diabetic macular edema.

### 5.1. Neuroprotectants for Early DR

Somatostatin, a neuropeptide known for its inhibitory effects on endocrine and exocrine hormone secretion, has been shown to improve neuronal degeneration through topical administration [181,182] in diabetic animal models. However, findings from a multicenter, randomized, controlled clinical trial known as the EUROCONDOR study indicated that somatostatin eye drops did not show any neuroprotective effects in patients with early DR [183]. However, in a subpopulation of patients with pre-existing neuronal dysfunction—as determined by multifocal ERG—somatostatin appeared to ameliorate the progression of the neuronal dysfunction [183]. Although this study did not meet its primary endpoints, it suggests the importance of identifying biomarkers that can pinpoint subpopulations with pre-existing neuronal dysfunction in early DR [183]. To further investigate this, a case-control study from the placebo arm of the EUROCONDOR study has been performed [184]. Although the number of included patients was relatively small (n = 38), glial fibrillary acidic protein was identified as a useful biomarker for retinal neurodysfunction in patients with early DR [184]. Another study examines N-epsilon-carboxy methyl lysine (CML), laminin P1, and symmetric dimethylarginine in the serum of 341 patients of the EUROCONDOR study [185]. The study identified CML as a biomarker of neuronal dysfunction in early DR [185]. These findings are crucial as they pave the way for identifying appropriate candidates for early intervention with neuroprotective therapies aimed at combating DR.

CoQ10 (ubiquinone) is a crucial biochemical cofactor that forms a critical link in the electron transport chain in the inner mitochondrial membrane and is involved in the process of oxidative phosphorylation [186]. CoQ10 acts as an antioxidant and free radical scavenger, benefiting patients with diabetes by positively affecting the mitochondrial respiratory chain and via recoupling endothelial nitric oxide synthase [187]. Domanico et al. examined the antioxidative effect of CoQ10 and vitamin E in patients with nonproliferative DR (n = 68) and reported that antioxidants reduce the blood level of ROS and restores retinal thickness [188]. Furthermore, randomized, double-blind, and placebo-controlled Phase IIa studies indicate that CoQ10 and combined therapy (10 mg of lutein, 4 mg of astaxanthin, 1 mg of zeaxanthin, 180 mg of vitamin C, 30 mg of vitamin E, 20 mg of zinc, and 1 mg of copper) significantly improve various parameters, including the membrane fluidity of erythrocytes, ATP hydrolysis, fluidity of submitochondrial particles of platelets, serum levels of lipid peroxidation, total antioxidant capacity, catalase activity, and glutathione peroxidase activity in patients with nonproliferative DR [189,190]. Thus, CoQ10 and adjunctive antioxidant treatments demonstrate promising potential in improving mitochondrial dysfunction and oxidative stress in nonproliferative DR [189,190]. Further randomized Phase 3 clinical trials should be performed to examine the effect of CoQ10 on preventing the progression of early DR.

Citicoline eye drops have already been utilized in the clinical practice for patients with glaucoma (see Section 2), offering safety and efficacy as neuroprotectants for the retina and optic nerve. Because RGCs are affected in the early stage of diabetes [191,192,193,194,195], a 3-year pilot study with a prospective, randomized, and double-masked design was performed to examine the effect of citicoline and vitamin B_12_ eye drops on changes in function and morphology of retinas in early DR [196]. Although the number of patients was small, citicoline and vitamin B_12_ eye drops significantly restored visual function and neurodegenerative changes in early DR [196]. Furthermore, another same series of the pilot study indicates that citicoline and vitamin B_12_ eye drops significantly improved macular function, measured by multifocal ERG, in patients with early DR [197]. Further large-scale randomized clinical trials are warranted to examine the effect of citicoline eye drops in delaying the progression of early DR.

### 5.2. Existed Medical Drugs

DR is a complication arising from a systemic disease, diabetes mellitus. Thus, various medical drugs have been employed for patients with diabetes and DR. Among these, several show significant therapeutic effects on the development and progression of neurovascular abnormalities associated with DR. Fenofibrate, a peroxisome proliferator-activated receptor alpha (PPAR*α*), is a medical drug for hyperlipidemia. Two large, randomized clinical trials (ACCORD Eye study and FIELD study) indicate that fenofibrate significantly reduces the rate of progression of DR [198,199]. Several basic studies confirm the biochemical effects of fenofibrate on neurovascular abnormalities of DR. Interestingly, this sequence might appear counterintuitive—often referred to as “from bed to bench”. Fenofibrate decreases overexpression of ICAM-1, VEGF, and monocyte chemotactic protein-1, along with inhibiting hypoxia-inducible factor-1 and NF-*κ*B, thereby ameliorating vascular leakage in DR [200]. Furthermore, fenofibrate reduces overexpression of the extracellular matrix components fibronectin and collagen IV in endothelial cells and RPEs, thereby mitigating the blood–retinal barrier dysfunction [200,201,202]. Finally, fenofibrate reduces glial activation and retinal neuronal apoptosis, resulting in improved ERG parameters [203]. Taken together, these preclinical studies support clinical evidences by demonstrating that fenofibrate has both neuroprotective and vasoprotective effects against diabetic stress in early DR. Metformin, antihyperglycemic drug, also shows potential neuroprotective effects, primarily by enhancing mitochondrial function through the activation of AMP-activated protein kinase (AMPK) [204]. The AMPK/mammalian target of rapamycin signaling pathway is associated with the neuroprotective effect of metformin in patients with diabetes suffering from acute stroke [205]. Metformin shows protective effect on retinal cells against diabetic and oxidative stresses by suppressing toll-like receptor 4/NF-*κ*B [206]. A population-based cohort study indicated that metformin reduces the risks of developing nonproliferative and sight-threatening DR [207].

A wide array of therapeutic approaches have been explored to prevent the development and progression of early DR by protecting the neurovascular units. However, it is important to note that DR is a complication of a systemic disease—diabetes mellitus; various systemic conditions, such as hypertension, hyperlipidemia, or renal dysfunction, are partially associated with the microvascular abnormalities observed in DR. Therefore, researchers should not only focus on developing novel therapeutic approaches but also consider existing medications, such as glitazones, angiotensin-converting enzyme inhibitors, statins, metformin, and fenofibrate, all of which may have therapeutic effects on DR. As the number of patients at risk for sight-threatening DR is expected to rise in the coming years, further studies are needed to develop therapeutic approaches for neurovascular protection in early DR.

Table 1 shows a summary of clinical studies on neuroprotective and regenerative therapies for glaucoma, AMD, retinitis pigmentosa, and DR.

## 6. Future Perspectives

Neuroprotective and regenerative therapies are some of the options for preventing the progressive loss of vision with retinal and optic nerve diseases. However, once retinal neurons are lost, it is still difficult to recover visual function. Therefore, earlier detection and intervention of sight-threatening diseases are important for preventing blindness due to retinal and optic nerve diseases. For this purpose, finding the risk or predisposing factors for developing these diseases are crucial. In the case of glaucoma, IOP elevation, older age, myopia, systemic hypotension, migraine, sleep apnea syndrome, diabetes, family history, some systemic medications, and nonwhite race, have been reported as risk or predisposing factors [208,209]. Among them, only the elevation of IOP can be controlled for reducing risks of developing and progressing of glaucomatous optic neuropathy and thus, neuroprotective approaches are strongly eager for clinical management for patients with glaucoma. Optical coherence tomography (OCT) findings, including reduction in the retinal nerve fiber layer (RNFL) thickness, are useful for detecting glaucoma in subjects in whom it is suspected, because the RNFL thickness is reduced before visual field defects can be detected in most subjects with glaucoma suspects [208,209]. Screening of high-risk groups, via patient histories and examinations combined with OCT analyses and visual field tests, appears to be helpful for the early detection and intervention in patients with glaucoma.

The risk or predisposing factors of DR are undoubtedly higher hemoglobin A1c and longer duration of diabetes. Other common risk factors are hypertension, dyslipidemia, smoking, and higher body mass index [210]. Although other minor risk factors including genetic biomarkers, microRNA, and metabolomics are reported, further extensive investigation is still required for establishing a reliability of these factors for detecting DR [211]. Different from glaucoma, most risk factors of DR can be modified and controlled to prevent the progression of DR. Thus, early detection of diabetes and DR is more important than other retinal and optic nerve diseases. For screening, fundus and OCT examinations are conventionally required, but with recent advances in artificial intelligence and a deep learning algorithm, the FDA approved automated DR screening program, because the sensitivity and specificity in detecting DR reached 96.8% and 87%, respectively [212]. However, minimum screening examinations by ophthalmologists are still recommended by ADA and the International Council of Ophthalmology [211]. Taken together, the identification of disease risk or predisposing factors facilitates the earlier detection of glaucoma and DR followed by earlier initiation of interventions for these diseases.

Even though ophthalmologists make efforts to detect retinal and optic nerve diseases as early as possible, some patients with these diseases are found in advanced stages, because DR and glaucoma often progress without notification until advanced stages. The progression of GA and retinitis pigmentosa cannot be prevented completely, and thus, regenerative and cell replacement therapies are urgently required for patients with advanced stages of retinal and optic nerve diseases. For overcoming these limitations in clinical practice, tissue-engineering approaches have been investigated, especially for AMD [213] and retinitis pigmentosa [214]. Currently two- and three-dimensional tissue engineering approaches for cell- or organoid-derived sheet transplantation are available for AMD [213] and retinitis pigmentosa [214]. However, no current Phase III studies of stem cell therapies for AMD and retinitis pigmentosa have been performed. Human embryonic stem cells (MA09-hRPE) were subretinally transplanted in patients with dry AMD as Phase 1/2 trials, but there were four ocular treatment-emergent adverse events, i.e., two serious infections, one neurological event, two cases of squamous cell cancer, and one basal cell cancer (NCT01344993, NCT02463344) (references in Astellas Web page cannot be available now). Another human embryonic stem-cell-derived RPE cell transplantation Phases 1/2 trial (CPCB-RPE1) (NCT02590692) showed no serious adverse events and mild improvement in visual function in eyes treated with dry AMD compared to untreated eyes [215]. The first human trial of iPSC-RPE cell-sheet transplantation was performed and showed no adverse event in the first patient of neovascular AMD over one year [216], but in the second patient’s iPSCs, mutations were observed on the sex chromosome, and thus, the trial was forced to stop because of changes in the regulatory rules in Japan [216]. This study did not target patients with dry AMD but neovascular AMD, and thus, the iPSC-derived RPE cells on a poly lactic-co-glycolic acid scaffold transplantation Phases 1/2 trial is being conducted for patients with GA secondary to AMD (NCT04339764). Oncogenesis is an issue inherent in stem cell transplantation. Therefore, further extensive studies are required to establish the safe and tolerable tissue-engineering approaches for AMD and retinitis pigmentosa.

In conclusion, this narrative review presents a range of therapeutic approaches for the following four leading sight-threatening diseases in Japan: glaucoma, retinitis pigmentosa, DR, and AMD [1]. Each section highlights some translational research efforts that have moved from bedside to bench. Nevertheless, further Phase 3 randomized clinical trials are still required for many therapeutic approaches. Retinal and optic nerve diseases are difficult to treat in severe cases, and thus, hopefully, novel neuroprotective and regenerative approaches for sight-threatening retinal and optic nerve diseases will be established in the near future.

## Figures and Tables

**Figure 1 ijms-25-10485-f001:**
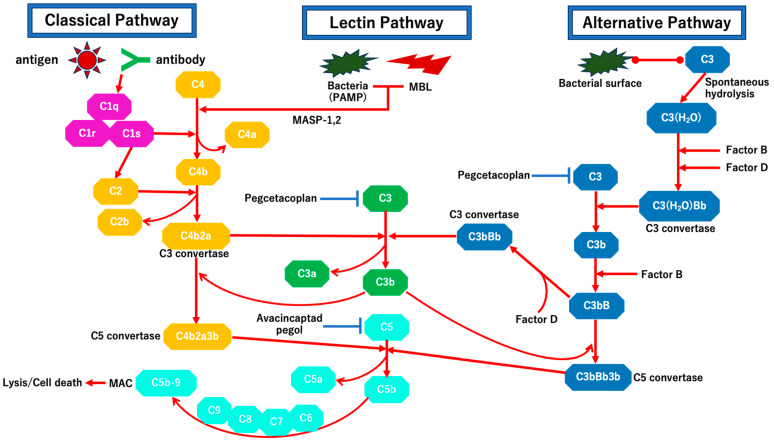
Schema of the complement pathways. There are three main pathways that activate complement, as follows: the classical pathway, the lectin pathway, and the alternative pathway. The classical pathway is activated when C1q binds to an antibody that is attached to an antigen. This binding activates C1r and C1s, which then cleaves C4 and C2 into C4a and C4b, and C2a and C2b, respectively. The lectin pathway is activated when mannose-binding lectin (MBL) binds to conserved pathogenic carbohydrate motifs (pathogen-associated molecular pattern; PAMP). This interaction triggers the cleavage of C4 into C4a and C4b under the activation of MBL-associated serine proteases (MASPs). Following the cleavage of C2 and C4, C4b and C2a make a complex, “C4b2a”, which is C3 convertase. The C3 convertase cleaves C3 into C3a and C3b. C3b forms a complex with C4b2a, “C4b2a3b”, which works as another C5 convertase, leading to the cleavage of C5 into C5a and C5b in both the classical and the lectin pathways. In the alternative pathway, C3 is spontaneously hydrolyzed into C3(H_2_O), which forms C3(H_2_O)Bb in the presence of Factors B and D. C3(H_2_O)Bb acts as C3 convertase, which cleaves C3 into C3b. If C3b binds to bacterial surface, Factor B and D interact to form a complex, C3bB followed by C3bBb, another C3 convertase. The C3bBb cleaves C3 into C3a and C3b, resulting in the formation of another C5 convertase, “C3bBb3b.” All three pathways culminate in the formation of the C5 convertases (C4b2a3b and C3bBb3b), ultimately leading to the formation of the membrane attack complex (MAC; C5b-9). At the same time, other factors generated during the MAC formation process (C3a, C4a, and C5a) act as anaphylatoxins. C5a acts as a chemokine that recruits neutrophils to inflammatory lesions. C3b binds to and opsonizes targets, facilitating phagocytosis and further amplifying complement activation in the alternative pathway. Pegcetacoplan (Syfovre) targets C3, while Avacincaptad pegol (Izervay) targets C5, inhibiting the MAC formation.

**Table 1 ijms-25-10485-t001:** Summary of neuroprotective regenerative therapies for glaucoma, AMD, retinitis pigmentosa, and DR.

Diseases	Drugs and/or Therapies	Remarks	Cites
Glaucoma	Bupropion	Norepinephrine–dopamine reuptake inhibitor (probably suppressing TNF production)	[32] (retrospective; positive)
	Memantine	Open-channel-blocking NMDA antagonist	[52] (RCT; negative)
	Citicoline	Intermediate product in the synthesis of phosphatidylcholine (approved by over 10 nations)	[65,66,67,68,69] [68] (RCT; positive) [69] (RCT; positive)
	Brimonidine	A_2_ agonist used as an IOP-lowering agent	[73] (RCT; positive)
	CNTF	Neurotrophic factors (activate the JAK/STAT signaling pathway)	[85] (Phase 1; CNTF implants) Phase II is underway (NCT02862938)
	Coenzyme Q10	Cofactor in the electron transport chain	[95] (CoQ10 + vitamin E; improve retinal function)
	Nicotinamide	Precursor of nicotinamide adenine dinucleotide	[102] (Phase II; positive) [103] (RCT; positive)
AMD	Pegcetacoplan (Syfovre)	Pegylated C3 inhibitor peptide (approved by FDA)	[125] (RCT; positive)
	Avacincaptad pegol (Izervay)	Anti-C5 aptamer (approved by FDA)	[126] (RCT; positive) [127] (RCT; positive)
Retinitis pigmentosa	Voretigene neparvovec (LUXTURNA)	AAV2 vector containing human RPE65 cDNA with a modified Kozak sequence (approved by FDA)	[150] (RCT; positive)
	EDTT-101	Gene-editing therapy mediated by AAV5 vector containing a pair of CEP290-specific guide RNAs coupled with Cas9	[157] (Phase 1/2; no serious adverse events)
	SPVN06	Human-engineered AAV delivered via RdCVF gene (RdCVF is a neurotrophic factor secreted from rod photoreceptors)	Phase1/2 is underway (NCT05748873)
	iPSC-derived organoid implantation	Allogeneic iPSC-derived retinal organoid sheets were transplanted	[171] (jRCTa050200027) (no adverse events but no functional improvement) (the first study on the transplantation of stem-cell-derived organoids in human)
	Optogenetics	AAV vector encoding ChrimsonR fused to tdTomamto	[174] PIONEER study (open-label Phase1/2a study) (recovery of visual function in blind patients)
	Optogenetics	AAV-mediated delivery of multicharacteristic opsin (MCO-010) for patients with severe sight loss due to retinitis pigmentosa	Phase 2b RESTORE trial (NCT04945772); confirmed safety and efficacy of MCO-010
	Optogenetics	MCO-010 for patients with Stargardt disease	Phase 2 STARLIGHT trial (NCT05417126); confirmed safety and efficacy of MCO-010
	Optogenetics	AAV-mediated ChronosFP (BS01) delivery into RGCs in retinitis pigmentosa	Phase 1/2 trial (NCT04278131); underway
	Optogenetics	AAV-mediated RST-001 delivery in patients with retinitis pigmentosa	Phase 1/2a trial (NCT02556736); ongoing
DR	Somatostatin and Brimonidine	Somatostatin (a neuropeptide inhibiting endocrine and exocrine hormone secretions) and brimonidine eye drops in patients with early DR	[183] RCT (The EUROCONDOR study); negative, except for subgroups with pre-existing neuronal dysfunction
	Citicoline	Intermediate product in the synthesis of phosphatidylcholine	[196] 3-Year pilot study (citicoline + vitamin B_12_ eye drops restored visual function and neurodegenerative changes)
	Fenofibrate	PPAR*α*, a medical drug for hyperlipidemia	[198] RCT (ACCORD study); positive [199] RCT (FIELD study); positive
	Metformin	Biguanide antihyperglycemic agent	[207] Population-based cohort study; reduced the risks of sight-threatening DR

## Data Availability

Not applicable.

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
