# Peer review of "Translational Research and Therapies for Neuroprotection and Regeneration of the Optic Nerve and Retina: A Narrative Review"

_ijms, 2024, doi:10.3390/ijms251910485_

Round 1

Reviewer 1 Report

Comments and Suggestions for Authors

This review manuscript summarizes the therapeutic approaches for four retina-related diseases: glaucoma, AMD, retinitis pigmentosa, and diabetic retinopathy, and provides some translational and clinical perspectives of future development. Here are some suggestions that might help improve the paper:

1. The review consists of an introduction and four extensive sections discussing different ocular diseases. To improve readability and help the audience navigate, consider adding subsection titles within each section, such as "Drug Treatment", "Gene Therapy", "Stem Cell Therapy", etc.

2. The author summarizes various signaling pathways and molecules that are under investigation for ocular diseases. Including a table that outlines the current status of clinical trials and their outcomes would be highly beneficial for readers and future drug development.

3. The title highlights both neuroprotection and regenerative therapies for retinal diseases. However, retina regenerative medicine is only sparsely discussed compared with the neuroprotection approaches. To provide a more balanced overview, please consider expanding the discussion on regenerative therapies, such as stem cell therapy and tissue engineering approaches. Here are some references that might be helpful:

Nair DSR, Seiler MJ, Patel KH, Thomas V, Camarillo JCM, Humayun MS, Thomas BB. Tissue Engineering Strategies for Retina Regeneration. Appl Sci (Basel). 2021 Mar;11(5):2154. doi: 10.3390/app11052154. Epub 2021 Feb 28. PMID: 35251703; PMCID: PMC8896578.

Wu A, Lu R, Lee E. Tissue engineering in age-related macular degeneration: a mini-review. J Biol Eng. 2022 May 16;16(1):11. doi: 10.1186/s13036-022-00291-y. PMID: 35578246; PMCID: PMC9109377.

Author Response

Editorial Office

To the Editor

We submit our revised manuscript titled,

Translational researches and therapies for neuroprotection and regeneration of the optic nerve and retina: A narrative review

to be considered for publication in International Journal of Molecular Sciences. Our responses to the reviewers’ comments are presented below.  We would appreciate your kind review again.

Sincerely,

Toshiyuki Oshitari, MD, Ph.D.

Department of Ophthalmology and Visual Science

Chiba University Graduate School of Medicine

Inohana 1-8-1, Chuo-ku, Chiba 260-8670, Chiba, Japan.

TEL:81-43-226-2124

FAX:81-43-224-4162

e-mail:[email protected]

Reviewer 1

  1. To improve readability and help the audience navigate, consider adding subsection titles….

Answer. According to your suggestion, we have added subsection titles within each section.

  1. Including a table that outlines the current status of clinical trials…

Answer. Thank you for your suggestion. We have added a new table (Table 1) that outlines the current status of clinical trials and their outcomes.

  1. Please consider expanding the discussion on regenerative therapies…

Answer. We have added the discussion on regenerative therapies especially tissue engineering approaches in the “future perspective” (the last) section. According to your suggestion, we have cited your recommended references in the text.

Reviewer 2 Report

Comments and Suggestions for Authors

This is a very interesting manuscript from a clinical standpoint, and the thought behind this paper is very good. The introduction is well-written, but the conclusion could be improved. The article provides good insight into the neuroprotective and regenerative therapeutic strategies for the clinical management of neurodegenerative and neurovascular conditions of the posterior segment of the eyes, such as glaucoma, retinitis pigmentosa, age-related macular degeneration, and diabetic retinopathy. Citicoline exerts a neuroprotective effect on damaged retinal neurons, and the author highlighted that it could be a promising neuroprotective therapy for glaucomatous optic neuropathy. Furthermore, the author highlighted the clinical utility of citicoline in patients with diabetic retinopathy due to its neuroenhancement and neuroprotective effects. The dysregulation of the complement system plays a significant role in the pathogenesis of age-related macular degeneration, and the author did a great job discussing the role of complement pathway inhibitors as an emerging therapeutic strategy in slowing down the progression of geographic atrophy in age-related macular degeneration.

Major issues

Although the author provided an excellent descriptive figure of the complement system, including complement pathway inhibitors, it would be interesting for readers if the author could provide a self-explanatory diagram for sections 2, 3, 4, and 5. This might help readers grasp the mechanism of action and clinical utility of the agents described in this manuscript. A table could also be a good alternative.

Lines 116 – 120: The cited reference does not support the claim made in this sentence. Please provide the appropriate source for this sentence.

Minor issue

Line 368: Render “EYE” as “EYS”. This should apply to the rest of the text.

Epidemiology was adequately discussed for the four conditions highlighted in this manuscript, but the discussion of the risk or predisposing factors for these conditions needs to be balanced. The revised version should adequately discuss the risk or predisposing factors for glaucoma and diabetic retinopathy.

Author Response

Editorial Office

To the Editor

We submit our revised manuscript titled,

Translational researches and therapies for neuroprotection and regeneration of the optic nerve and retina: A narrative review

to be considered for publication in International Journal of Molecular Sciences. Our responses to the reviewers’ comments are presented below.  We would appreciate your kind review again.

Sincerely,

Toshiyuki Oshitari, MD, Ph.D.

Department of Ophthalmology and Visual Science

Chiba University Graduate School of Medicine

Inohana 1-8-1, Chuo-ku, Chiba 260-8670, Chiba, Japan.

TEL:81-43-226-2124

FAX:81-43-224-4162

e-mail:[email protected]

Reviewer 2.

  1. It would be interesting for readers…. A table could also be a good alternative.

Answer. According your suggestion, we have added a new table (Table 1) that outlines the current status of clinical trials and their outcomes.

  1. Lines 116-120: The cited reference does not support the claim made in this sentence…

Answer. Thank you for your suggestion. We have replaced a new reference (No. 34).

  1. Line 368: Render “EYE” as “EYS”.

Answer. Thank you for your suggestion. We have changed “EYE” into “EYS” in the text.

  1. The discussion of the risk or predisposing factors for these conditions needs to be balanced…

Answer. According to your suggestion, we have added the discussion of the risk or predisposing factors for glaucoma and diabetic retinopathy in the last “future perspectives” section.

Round 2

Reviewer 2 Report

Comments and Suggestions for Authors

The author has addressed my concerns with the original manuscript.